# THz-TDS Techniques of Thickness Measurements in Thin Shim Stock Films and Composite Materials

Kwang-Hee Im [1,*], Sun-Kyu Kim [2], Young-Tae Cho [3], Yong-Deuck Woo [1] and Chien-Ping Chiou [4]

1 Department of Automotive Engineering, Woosuk University, Wanju-kun, Jeonbuk 55338, Korea; wooyongd@woosuk.ac.kr
2 Division of Mechanical System Engineering, Jeonbuk National University, Jeonju, Jeonbuk 54896, Korea; sunkkim@jbnu.ac.kr
3 Department of Basic Science, Jeonju University, Jeonju, Jeonbuk 55069, Korea; choyt@jj.ac.kr
4 Aerospace Engineering and Center for Nondestructive Evaluation, Iowa State University, Ames, IA 50011, USA; cchiou@iastate.edu
* Correspondence: khim@woosuk.ac.kr; Tel.: +82-63-290-1473

**Abstract:** Terahertz wave (T-ray) scanning applications are one of the most promising tools for nondestructive evaluation. T-ray scanning applications use a T-ray technique to measure the thickness of both thin Shim stock films and GFRP (glass fiber-reinforced plastics) composites, of which the samples were selected because the T-ray method could penetrate the non-conducting samples. Notably, this method is nondestructive, making it useful for analyzing the characteristics of the materials. Thus, the T-ray thickness measurement can be found for both non-conducting Shim stock films and GFRP composites. In this work, a characterization procedure was conducted to analyze electromagnetic properties, such as the refractive index. The obtained estimates of the properties are in good agreement with the known data for poly methyl methacrylate (PMMA) for acquiring the refractive index. The T-ray technique was developed to measure the thickness of the thin Shim stock films and the GFRP composites. Our tests obtained good results on the thickness of the standard film samples, with the different thicknesses ranging from around 120 μm to 500 μm. In this study, the T-ray method was based on the reflection mode measurement, and the time-of-flight (TOF) and resonance frequencies were utilized to acquire the thickness measurements of the films and GFRP composites. The results showed that the thickness of the samples of frequency matched those obtained directly by time-of-flight (TOF) methods.

**Keywords:** terahertz waves; refractive index; thickness measurement; Shim stock films; composite materials; reflection mode

## 1. Introduction

Terahertz waves (T-ray) have recently been utilized for technical applications [1]. Along with the recent progress of T-ray technology and monitoring instruments, defect inspection methods have emerged based on the electronic spectrum. Moreover, the T-ray has a relatively higher resolution. In addition, the T-ray has led to advanced progress for spectroscopic monitoring in security areas, food inspection, water, the mechanical field, and materials. Terahertz time domain spectroscopy (THz-TDS) has been utilized to inspect various delamination or foreign materials in advanced non-contact composites. THz-TDS is based on photoconductivity, and this depends on low-cycle formations with the utilization of a photoconductive antenna (Femtosecond (10–15 s) laser) [2].

It is possible to create THz waves in less than a pico-second. Therefore, detection techniques using a high signal-to-noise (S/N) ratio are available, which affects the broad bandwidth. A temporary change in the T-ray emitter occurs due to the resistance of the photoconductive switch on the T-ray timescale [3,4]. In addition, another method, known as optical anisotropic conversion or optical mixing, can be utilized along with

two continuous wave (CW) lasers [5]. When these two lasers are mixed, a beating is generated, and this beating can modulate the conductance of the photoconductive switch using the terahertz differential frequency [6]. Continuous-wave terahertz (CW-THz) can be obtained using this method. In some cases, a T-ray image can also show the chemical components of a target material [7]. Owing to these characteristics, the T-ray image has attracted significant attention. The T-ray image has commercial applications in various fields, including humidity analysis, quality management of plastic products, and packing inspection (monitoring) [8–10].

Owing to its broad utilization and far-ranging applications, the THz-TDS techniques could have the possibility to become a portable THz image. This approach is composed of two sections, which both involve the use of the T-ray. First, the importance of fiber-reinforced plastics (FRP) in the space and civil aviation fields is generally well known, and the FRP-laminated plate is widely used. In addition, the waveforms of terahertz pulses in the TDS mode have a strong resemblance to those of ultrasonic tastings. Regarding wave propagation concepts such as time of flight (TOF), transmission and reflection coefficients, refraction and diffraction are common to both waves. However, there are also fundamental differences when materials are probed with terahertz radiation, an electromagnetic wave, and with ultrasound, a mechanical wave [11]. In order to measure the thickness of a specimen using conventional ultrasonic waves, a couplant medium is always required, which makes the ultrasonic waves easily propagated. In the case of using air as a couplant medium, selecting ultrasonic frequency is narrowly ranged; thus, there is a limitation to measuring thinner samples. Therefore, due to the couplant medium, the factors affecting the accuracy of the measurements should be considered such as attenuation, diffraction, and dispersion of the samples [12]. By the way, the terahertz wave used in this study requires no couplant medium and is utilized under the mode of noncontact. Thus, the terahertz wave could make better reproducibility of data produced and also a higher frequency could be selected, which could bring the stronger measurement of thickness in case of thin samples.

The other is composed of the refractive index ($n$), the electrical conductivity of fiber-embedded epoxy matrix composite material, and the measurement of T-ray thickness for both glass fiber-reinforced plastics (GFRP) and thin Shim stock films, which are produced as a standard sample with an arbitrary thickness (ranging from tens of μm to hundreds of μm) [13]. Thus, the thicknesses for both GFRP and thin Shim stock films are measured using T-ray technology. Carbon fiber-reinforced plastics (CFRP) are conductive, but epoxy matrix is non-conductive [14,15]. However, the carbon fiber of the CFRP-laminated plate has conductivity, enabling the T-ray characteristics evaluation of glass fiber and carbon fibers [16].

In this study, the results of the experiment on the T-ray were obtained based on the non-destructive evaluation methods using FRP composite materials. In addition, the correlation was performed between the fiber direction and the E-field of the GFRP composites and the CFRP composite-laminated plate according to the refractive index measurement technique, which shows the properties of various materials and the existence of conductivity. A new numerical method of measurement of refractive indexes in reflection and transmission modes was proposed. In addition, we performed a fundamental experimentation and brought a simple testing procedure for acquiring the thickness of samples as an existing NDE method. Here, the measured thickness and the reference thickness of the Shim stock films, which had a standard reference thickness, were compared. In addition, the thicknesses of the GFRP composites with non-conductivity were measured.

Therefore, a difference in the time-of-flight (TOF) was utilized to measure the thickness of the GFRP composites using the T-ray. The effectiveness of a T-ray examination was successfully evaluated by comparing and reviewing the specimens using the resonance frequency.

## 2. Fundamental Theory

### 2.1. Measurement of Refractive Index

Using the refractive index measurement technique, the reflection mode was applied in the time domain of the T-ray, and the refractive index was induced by picking up a signal reflected through the specimen. The progress direction of the T-ray signal is shown in Figure 1. Here, $T$ is the transmitter of T-ray and $R$ is the receiver of T-ray.

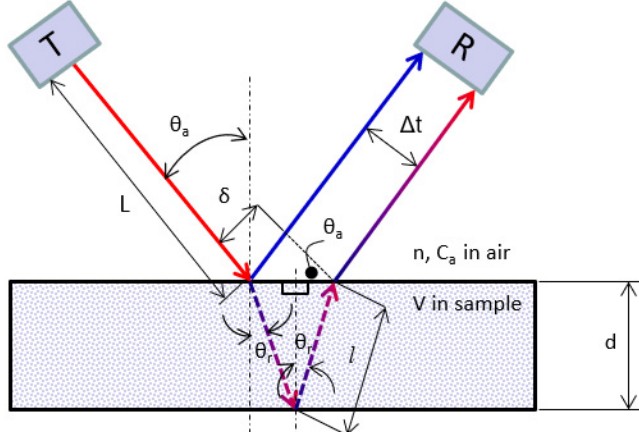

**Figure 1.** Diagram showing the geometry of the reflection mode [2].

The refractive index can be obtained by calculating the time of the T-ray, which is reflected from the terahertz pulsed emitter's arrival at the pulsed receiver and the time-of-flight (TOF) when the T-ray passes through a specimen of a certain thickness [1].

This reflection mode obtains the refractive index by calculating each length of optical fiber reflected on the top and bottom of the specimen in the T-ray time-of-flight (TOF). Figure 1 shows the shape and path of T-ray. At first, if it is assumed that T-ray is projected on the specimen vertically, a time difference ($\Delta t$) can be obtained as follows:

$$\Delta t = \frac{2d}{v} \tag{1}$$

In consideration of the path of the oblique T-ray and the shape delay time in the reflection mode, as shown in Figure 1, a time difference ($\Delta t$) between the surface-reflected wave and back-reflected wave on the specimen can be obtained as follows:

$$\Delta t = \frac{2l}{v} - \frac{\delta}{C_a} \tag{2}$$

Here, $l = \frac{d}{cos\theta_r}$, $\delta = 2l sin^2\theta_a = 2\frac{d}{cos\theta_r} sin^2\theta_a$, $C_a$ is the velocity in air, $d$ is the sample thickness, $v$ is the sample velocity, $\theta_a$ is the angle of inclination in the reflection mode, $\theta_r$ is the refractive angle in the sample, and $n$ is the refractive index. When the shape delay time and the path of oblique T-ray are traced, both the time difference ($\Delta t$) and resonance frequency ($\Delta f$) can be expressed as follows [13]:

$$\Delta t = \left( \frac{2d}{v\,cos\theta_r} - \frac{\delta}{C_a} \right) = \frac{2d}{cos\theta_r}\left( \frac{1}{v} - \frac{sin^2\theta_a}{C_a} \right) \tag{3}$$

$$\Delta f = \frac{1}{\left( \frac{2d}{v\,cos\theta_r} - \frac{\delta}{C_a} \right)} = \frac{1}{\frac{2d}{cos\theta_r}\left( \frac{1}{v} - \frac{sin^2\theta_a}{C_a} \right)} \tag{4}$$

Here, $l$ is the refracted length in the sample, $d$ is the thickness of specimen, $v$ is the velocity in the specimen, $C_a$ is the velocity in the air, $l$ is the refracted length in the sample, $\delta$ is the skip length of refractive waves in the sample, $\theta_r$ is the refraction angle in the

specimen, and $\theta_a$ is the refraction angle in the air. The refractive index, which is one of the electromagnetic properties, can be calculated by following the steps above.

The refractive index can be obtained with the approximate solution as follows:

$$n^4 - An^4 - A\sin^2\theta_{p1} = 0 \tag{5}$$

where $d$ is the sample thickness, $V_{air}$ is the light speed in air, and $V_s$ is the light speed in the sample. $\Delta t$ $(T)$ is the difference time between with sample and without sample, and $A = \frac{T^2 V_{air}^2}{4 d_2^2}$. Here, assuming that the normal reflection mode is vertical on the sample, the refractive index ($n$) should be $v\Delta t/2d$. However, this reflection mode was composed with some angles. Therefore, a correction factor needs to be considered to obtain a better solution, as shown in Equation (5).

*2.2. Measurement of Refractive Index*

In through-transmission mode, the index of refraction ($n$) can be calculated using the following equation, according to [2]:

$$\therefore n = 1 + \frac{\Delta t \ v_{air}}{d} \tag{6}$$

where $\Delta t$ is the time cap between with sample and without sample, $d$ is the sample thickness, $V_{air}$ is the light speed in air, and $L$ is the distance between the pulsed emitter and pulsed receiver.

**3. Experiment System and Measurement**

*3.1. Measurement System*

Figure 2 shows a photo of the THz-TDS system, which is a non-destructive testing device. This system is used to collect the material characteristics and scan the image of the specimen. The T-ray system used in this study was produced by TeraView Ltd. Cambridge in the United Kingdom. This system was composed of the time domain spectroscopy (TDS) pulse device and the frequency domain continuous wave (CW) device. It was composed of TDS technologies for generating, controlling, and searching terahertz pulses. The THz-TDS system can obtain an image and improve data acquisition, and its unique structural characteristics for manipulating the T-ray have a direct influence on the image production experiment. This TDS system had a frequency range from 50 GHz to 4 THz, and the delay time reached up to 300 ps. The T-ray beam was concentrated on the focal distances of 50 mm and 150 mm, and the full width at half-maximum (FWHM) values were 0.8 mm and 2.5 mm, respectively. This TDS device can be set for measuring the penetration or reflection (small-angle pitch-catch). The frequency range of the CW device was between 50 GHz and 1.5 THz. The focal distances of the CW device were also 50 mm and 150 mm. The TDS and CW devices were connected to each other through the optical fiber. Figure 3 shows the schematic diagram of the T-ray system.

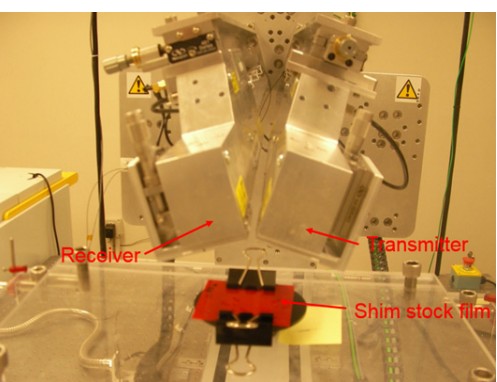

**Figure 2.** A photo of the THz-TDS system for imaging and measuring material parameters.

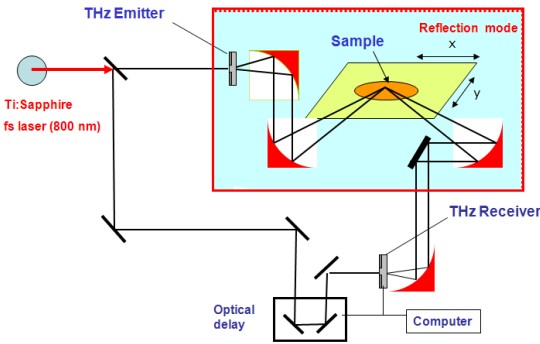

**Figure 3.** Overview of the THz measurement method [2].

### 3.2. Measurement Method

Figure 3 exhibits the T-ray measurement system, which demonstrates the reflection mode. When the test was carried out in this system, the T-ray was created from the emitter and sent to the receiver. At this time, the test was carried out by matching the focal point of the emitter and the receiver with the desired specimen. Then, the angle of inclination of the T-ray lens was determined as 16.6°. Figure 4 shows the Shim stock films and GFRP composites. The thicknesses of the Shim stock films were 0.127 mm, 0.254 mm, 0.381 mm, 0.508 mm, and 0.762 mm, and the thicknesses of GFRP composites were 2.02 mm, 3.08 mm, 5.74 mm, and 5.92 mm, respectively. Figure 5 shows typical A-scan data, which is the reflection mode of the GFRP composites of the T-ray. The thickness of the specimen was 3.0 mm.

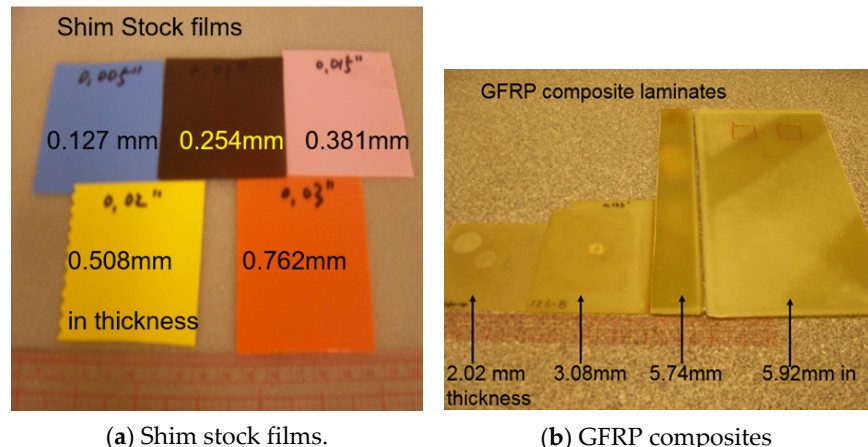

(**a**) Shim stock films.       (**b**) GFRP composites

**Figure 4.** Samples of (**a**) Shim Stock films and (**b**) GFRP composites with various thicknesses.

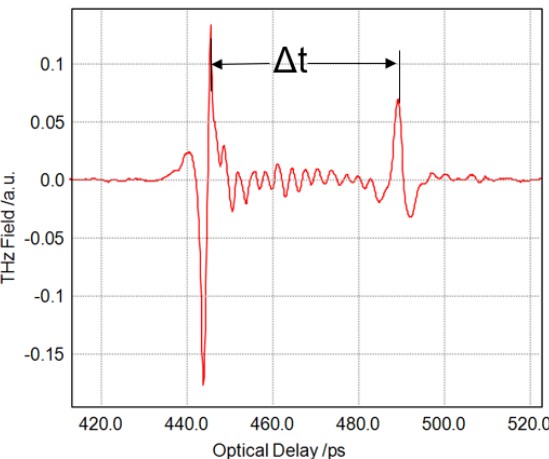

**Figure 5.** T−ray pulses from the transmitted GFRP composites ($n = 2.13$ $\Delta t = 42.7$ ps, $t = 3.0$ mm).

## 4. Results and Discussion

### 4.1. Measurement of Terahertz Refractive Index

To measure the T-ray parameter which indicated the material properties, the THz pulse was obtained from the Shim stock films and GFRP composites in the reflection mode. Figure 5 clearly shows the time difference ($\Delta t$) between the surface and the back of the GFRP composites in the reflection mode. GFRP was used for the specimen, and the thickness of specimen was approximately 3.0 mm. The time difference ($\Delta t$), which was obtained according to the thickness of the specimen, was 42 ps. Therefore, the optical time difference was calculated using the reflection mode, which is a measurement technique used to obtain the refractive index. The optimal time difference was calculated using Equation (4). In addition, the Shim stock films, GFRP composites, PMMA, and fused quartz specimens were measured under the reflection mode method, as shown in Table 1. When the results were compared with those from the previous references, only a difference within ±1% was found [1,6].

**Table 1.** Averaged THz refractive indices of the individually studied materials.

| Materials | Refractive Index (*n*) * | Refractive Index (*n*) |
|---|---|---|
| | | Reflection Mode |
| PMMA | 1.60 ± 0.08 | 1.59 ± 0.07 |
| Shim Stock films | - | 1.52 ± 0.03 |
| Fused quartz | 1.95 ± 0.05 | 1.94 ± 0.09 |
| GFRP | - | 2.17 ± 0.05 |

* Data in References [6,11].

Here, the reflection mode measurement techniques of the terahertz were performed in one direction, and experiments were carried out considering various aspects. In addition, since the measurement methods and the characteristics of the GFRP composites and Shim stock films were different, it was difficult to compare them with the previous data.

### 4.2. Electric Field Evaluation of Carbon Fiber

Unlike non-conductive materials, the T-ray has limited penetrating power against conductive materials [17]. At first, the test was carried out by applying the T-ray GFRP composites composed of non-conductive materials and the CFRP composites composed of conductive materials partially. The CFRP composites are composed of carbon fiber with conductive and non-conductive resin. When the CFRP-laminated composite plate is observed with a microscope, it is composed of various fibers and resins that could affect conductivity significantly, so the quantitative characteristic evaluation of carbon fiber composite material of T-ray is necessary. According to the previous reference, the radial conductivity of carbon fiber is approximately three times larger in the case of the electrical conductivity on the carbon fiber axis. The CFRP composites are composed of unidirectional composites, and the conductivity of the CFRP laminated plate composed of various lamination layers is affected. A transverse (vertical to the fiber axis) conductive generator depends on the fiber contact that occurs between adjacent fibers. Studies regarding the electrical conductivity of carbon fiber composite material are scarce. In some references, researchers have reported that the value of longitudinal conductivity ($\sigma l$) ranges between $1 \times 10^4$ s/m and $6 \times 10^4$ s/m, and the value of transverse conductivity ($\sigma t$) ranges between approximately 2 s/m and 600 s/m, which is much wider [18].

The transverse conductivity value of the laminated plate using the unidirectional Prepreg sheet varies significantly according to the production process and the quality of the laminated plate. The plane conductivity on the flowing current, while forming the $\theta$ angle with the fiber axis in the unidirectional CFRP composites, is given as follows [19]:

$$\sigma = \sigma l cos^2 cos\theta + \sigma t sin^2\theta \tag{7}$$

Since it is significantly higher than the longitudinal conductivity of the fiber ($\sigma l \gg \sigma t$), the T-ray which penetrates the unidirectional CFRP composites significantly varies according to the angle between the electric field vector and the axis of the carbon fiber. When the electric field of the T-ray is parallel to the axial direction of carbon fiber, the conductivity becomes the largest and the penetrating power becomes the smallest. On the contrary, when the electric field vector is perpendicular to the axis of fiber, the conductivity becomes the smallest and the penetrating power becomes the largest. The surface depth of the unidirectional oriented CFRP composites on the T-ray using the value of 10 s/m is 0.2 mm in 1 THz and 0.5 mm in 0.1 THz when the direction of electric field is vertical to the fiber axis. The effect of the penetrating power on the angle in the 24-ply unidirectional CFRP composite-laminated plate was experimentally evaluated using the CW terahertz device.

Figure 6 exhibits the amplitude profile of the penetrating power of both the GFRP and CFRP composites by the function of angle under the frequency of 0.1 THz. The amplitude profile of power was obtained, with values ranging from 0° to 90° for both the GFRP composites and CFRP composites. Notably, in the case of the GFRP composites, there was no change in the amplitude profile. However, the CFRP composites showed a higher amplitude of penetrating power at 90°, although they showed almost no amplitude of penetrating power at 0°. When the measurement was made, the GFRP composites were not dependent on any angle, but the CFRP composites were dependent on the angle of the carbon fibers.

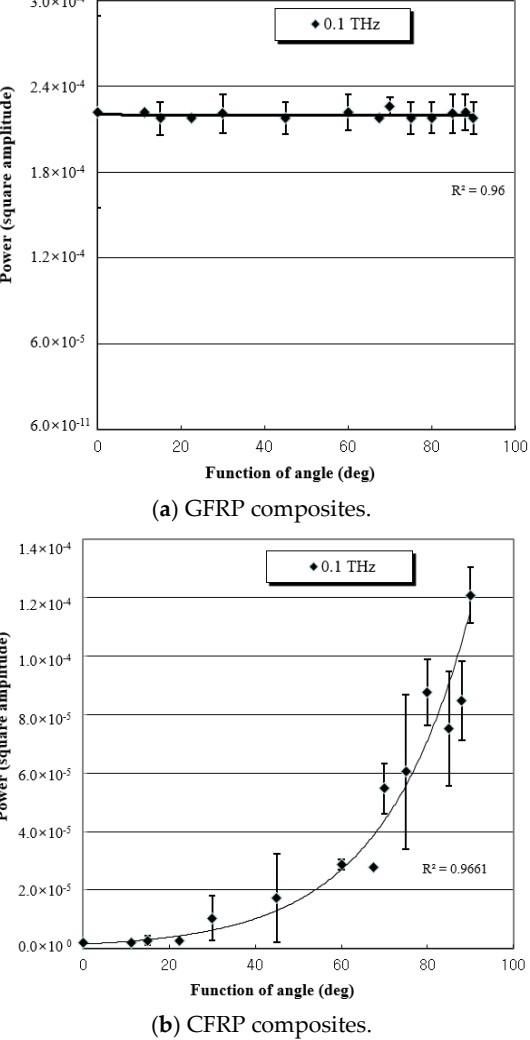

(**a**) GFRP composites.

(**b**) CFRP composites.

**Figure 6.** Amplitude profile of the penetrating power of both the GFRP and CFRP composites by the function of angle (a 24−ply glass and carbon composites).

### 4.3. Measurement of Thickness Using the Reflection Mode

The THz-TDS reflection mode was applied to measure the thickness for both the Shim stock films and GFRP composites with the one-side direction. Figure 7 exhibits the T-ray scan images of the thin Shim stock films. The thicknesses of the thin Shim stock films were 0.127 mm, 0.254 mm, 0.381 mm, 0.508 mm, and 0.762 mm. The values of the thicknesses were utilized as the standard samples of the films. Figure 7a exhibits the difference ($\Delta t$) in the time-of-flight (TOF), which indicates the difference between the surface and the back of the Shim stock films. Figure 7b represents Figure 7a as the FFT domain, and $\Delta f$ refers to the resonance frequency, which is correlated with the thickness of the thin Shim stock films. Here, $\Delta t$ is the difference time in the time-of-flight (TOF). Namely, $1/\Delta t$ should be $\Delta f$. Here, the example thickness in the thin Shim stock film was 0.381 mm.

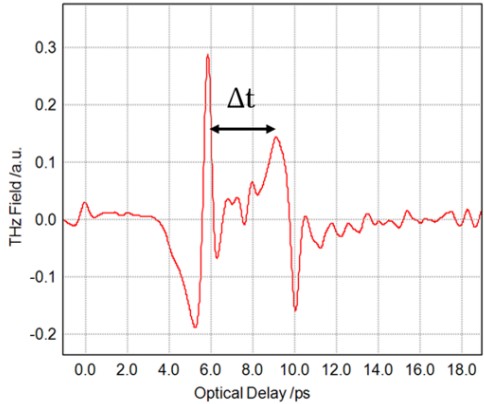

(**a**) A−scan image.

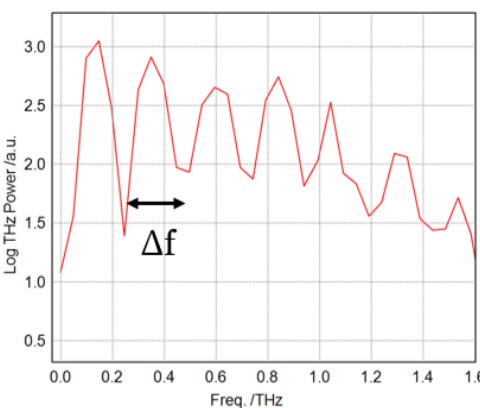

(**b**) Frequency−domain signal.

**Figure 7.** A TOF and FFT image of the thin Shim stock films under the reflection mode (0.381 mm in thickness).

Table 2 shows the comparison of the TOF difference ($\Delta t$) of the thin Shim stock films, resonance frequency ($\Delta f$), and T-ray measurement and reference thickness. Figure 8 shows the data after the T-ray scanning GFRP composites. The thicknesses of the GFRP composites were 2.02 mm, 3.08 mm, 5.74 mm, and 5.92 mm. The values of the thicknesses were used as the standard thickness of the samples.

Figure 8a shows the difference ($\Delta t$) in the time-of-flight (TOF), which indicates the difference between the surface and the back of the GFRP composites. Figure 8b represents Figure 8a as the FFT domain, and $\Delta f$ exhibits the resonance frequency, which is related to the thickness of the GFRP composites. Here, $\Delta t$ is the TOF difference. Namely, $1/\Delta t$ should be $\Delta f$. The thickness of the GFRP composites was 2.02 mm.

Table 3 shows the comparison of the TOF difference ($\Delta t$) of the GFRP composites, resonance frequency ($\Delta f$), T-ray measurement, and reference thickness.

**Table 2.** Measurements of the Shim stock films with various thicknesses using the THz techniques.

| Sample No. | Delay Time ($\Delta t$, ps) | Resonance Frequency ($\Delta f$) | T-ray Measurement (mm) | Reference Thickness (mm) | Others |
|---|---|---|---|---|---|
| 1 | 1.322 | 0.750 | 0.137 | 0.127 | Shim stock Co., Ltd. (Edenvale, South Africa) |
| 2 | 2.551 | 0.392 | 0.250 | 0.254 | |
| 3 | 3.846 | 0.260 | 0.396 | 0.381 | |
| 4 | 5.881 | 0.170 | 0.539 | 0.508 | |
| 5 | 8.600 | 0.115 | 0.789 | 0.762 | |

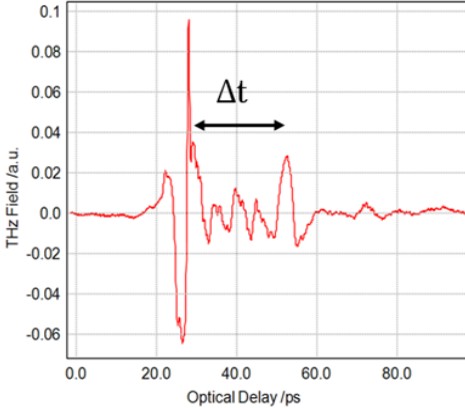

(**a**) A−scan image.

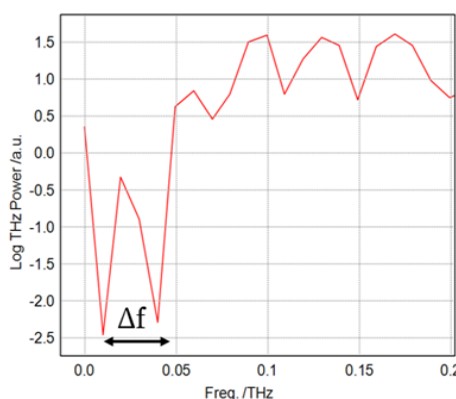

(**b**) Frequency−domain signal.

**Figure 8.** A TOF and FFT image of the GFRP composites under the reflection mode (2.02 mm in thickness).

**Table 3.** Measurements of the GFRP composites with various thicknesses using the THz techniques.

| Sample No. | Delay Time ($\Delta t$, ps) | Resonance Frequency ($\Delta f$) | T-ray Measurement (mm) | Reference Thickness (mm) | Others |
|---|---|---|---|---|---|
| 1 | 24.480 | 0.041 | 2.000 | 2.020 | Shim Stock Co., Ltd. |
| 2 | 50.000 | 0.020 | 3.180 | 3.080 | |
| 3 | 75.130 | 0.013 | 5.600 | 5.740 | |
| 4 | 83.30 | 0.012 | 5.920 | 5.920 | |

*4.4. Relation between Nominal Thickness and Thickness Measured from T-ray Techniques*

The Shim stock films and GFRP plates with non-conductivity were not dependent on the direction of the T-ray, so the measurement was possible. In addition, the T-ray reflection mode which could enable the measurement in one direction was adopted. Figure 9

exhibits the comparison between the nominal thickness in the thin Shim stock films and the thickness measured using the T-ray. As shown in Figure 9, the thicknesses of the thin Shim stock films were 0.127 mm, 0.254 mm, 0.381 mm, 0.508 mm, and 0.762 mm. The thickness of the thin Shim stock films was shown in a straight, solid line. This line shows the proportional relation with the standard thickness. Here, — represents the nominal thickness; □ represents the measured data in the case of the measurement, assuming that the T-ray was vertical to the specimen; and △ represents the measured data in the case of the inclined T-ray. Here, to effectively obtain the Refractive index (*n*), a suitable sample is the case with a thickness of several ones of mm. In case of the films, we did not prepare such a thicker sample. In this testing, this value is the average value of all the samples.

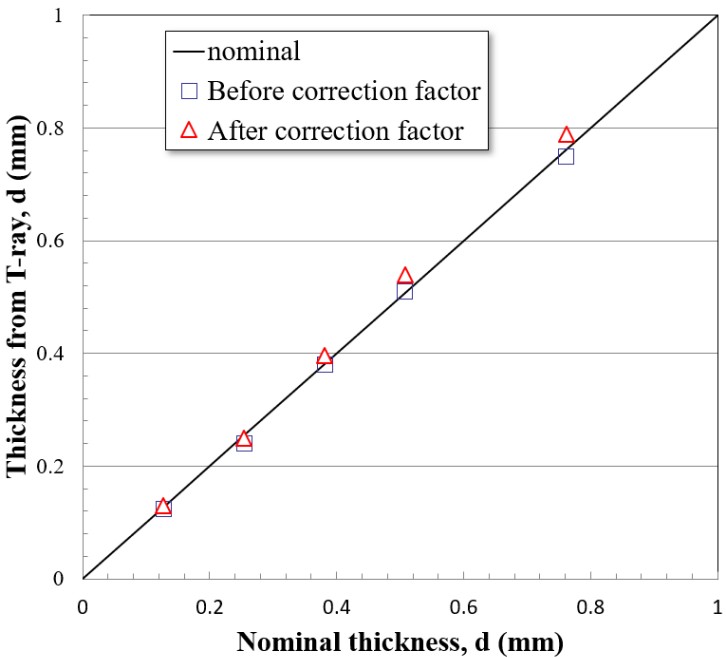

**Figure 9.** The relation between the nominal thickness and the thickness measured from the T-ray techniques in the Shim stock films.

Thus, the difference of data could be considered to be due to the average value of the refractive index to some degree. Even though there was a difference of ±2%, the results tended to be in agreement in the linear aspect.

Figure 10 shows the comparison between the nominal thickness of the GFRP composites and the thickness measured using the T-ray. The thicknesses of the GFRP composites were 2.02 mm, 3.08 mm, 5.74 mm, and 5.92 mm. In Figure 9, — represents the nominal thickness; □ represents the case of the measurement, assuming that the T-ray was vertical to the specimen; and △ represents the case of the inclined T-ray. Unlike the thickness at the microgram scale, the case of the inclined T-ray matched with the standard thickness at the millimeter scale. This can be attributed to the thickness of the specimen, the relatively small effect from the error, the strong received signal in electric field according to the fiber orientation of the GFRP composites, and the high penetration ratio of T-ray, enabling us to optimize a reception strong signal. Therefore, we found that it had potential reproducibility.

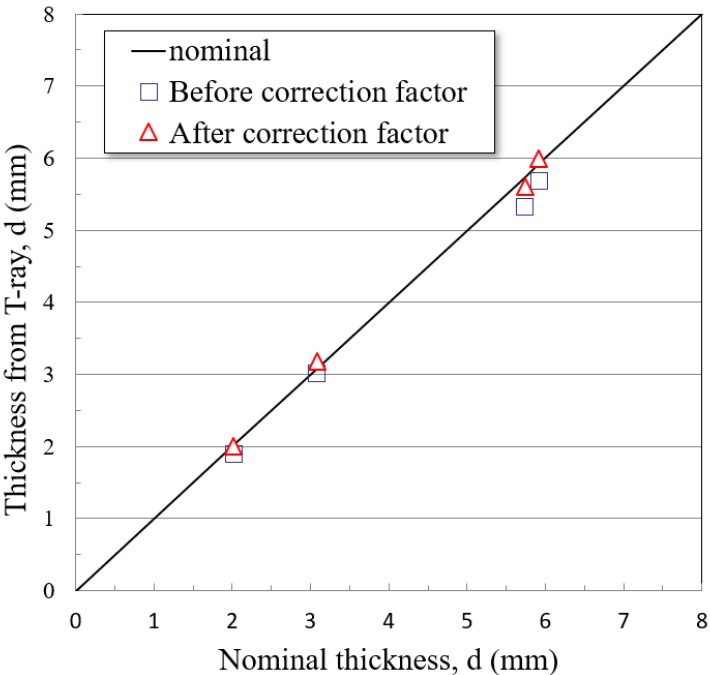

**Figure 10.** The relation between the nominal thickness and the thickness measured from the T-ray techniques in the GFRP composites.

## 5. Conclusions

In this approach, the refractive index measurement technique was established to calculate the material properties regarding the utilization of the T-ray for the non-destructive examination of Shim stock films and GFRP composites. In addition, the T-ray limitation in the energy penetrating power was discussed with respect to the conduction characteristics of the GFRP composites and the fiber lamination angle of CFRP plates. Possible THz-TDS techniques are summarized for measuring the thickness of the thin Shim stock films and GFRP composites as follows:

(1) It was possible to solve the refractive index of the thin Shim stock films and GFRP composites utilizing T-ray techniques under the reflection mode.

(2) The T-ray showed a constant level of penetrating power in the glass fiber class composites, which led to a very high penetration ratio and enabled the optimization of a strong reception signal. Therefore, it was found that it had potential reproducibility.

(3) The values of the measured thicknesses for both Shim stock films and GFRP composites were in agreement with those of the nominal thicknesses. The values were successfully measured through the correlation between the TOF cap and the resonance frequency under the reflection mode.

(4) We expect that the manufactured thickness measurement device using T-ray techniques may be very useful for non-destructive examinations in future applications in the advanced aerospace field.

**Author Contributions:** K.-H.I. suggested and designed the experiments; S.-K.K., Y.-T.C., and Y.-D.W. performed the experiments; C.-P.C. helped in the accomplishment of ideas and the administration of the experiments. The data were discussed and analyzed, and the manuscript was written and revised by all members. All authors have read and agreed to the published version of the manuscript.

**Funding:** This research was supported by the Basic Science Research Program through the National Research Foundation of Korea (NRF) (No. 2021R1I1A3042195) and also experimentally helped by the CNDE at Iowa State University, Ames, IA, USA.

**Institutional Review Board Statement:** Not applicable.

**Informed Consent Statement:** Not applicable.

**Data Availability Statement:** The data are contained within the article.

**Conflicts of Interest:** The authors declare no conflict of interest.

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
