# Peer review of "THz-TDS Techniques of Thickness Measurements in Thin Shim Stock Films and Composite Materials"

_applsci, doi:10.3390/app11198889_

Round 1
Reviewer 1 Report
Dear Authors,
1. Quality of content: In my opinion this is not enough for a publication. You buy a commercially available machine and put samples on it. Where is the novelty? Where is the innovation?
2. Quality of English: I highly recommend to involve a professional to improve grammar and style of the language and remove the various errors.
3. Quality of images: most images are blurry (Fig.4), makers show pixels (Fig. 2, 3, 9, 10), have too many leading zeros in the axes (Fig. 6) or show too small markers and inscriptions (Fig. 9 and 10).
Author Response
Please find a file attached herewith.
Thank you.
Kwang-Hee Im

Reviewer 2 Report
The authors use time domain THz technique to characterize materials properties such as thickness, refractive index and composition. This technique would be useful for thick, opaque/rough materials where visible light scatters too much. I support the publication with some minor comment listed below.
- References are largely missing, even in the introduction paragraph. In my opinion, every single sentence that gives a definitive statement beyond the data can support should be properly referenced.
- Equation 7 (line 238) is not correct. It's either a typo or some symbols are missing.
- The language/grammar can be improved throughout the manuscript.
Author Response

(The authors gave the same response as above.)

Reviewer 3 Report
The paper “THz-TDS Techniques of Thickness Measurements in Thin Shim Stock Films and Composite Materials)”, by Kwang-Hee Im, Sun-Kyu Kim, Young-Tae Cho, Yong-Deuck Woo and Chien-Ping Chiou, presents a study focused on T-ray examination of GFRP composite materials. The paper is well written, and the information is clearly presented.
The manuscript can be accepted for publication after Minor Revision.
- Figure 1: Is it an original representation? if not, please insert a reference.
- Please use the same Font size for all equations.
- Line 135: appears “…mode8…”, and I cannot see what does it means the superscript 8?
- Equation 6: what does it means the triangle formed from 3 points, that is placed in front of “n”...? Is it a mathematical explanation, or is by mistake?
- Figure 4: Image (a) it’s blurry…can be inserted another image which is clearer?
- Figure 5: Please insert a clearer graph.
- Line 200: inside table description- please correct “…material studied” in “ … studied material”.
- Figure 6, graph b: why does it appears big error bars at high angles?
- Line 386: here appears a square figure. Please remove it.
Author Response

(The authors gave the same response as above.)

Round 2
Reviewer 1 Report
Thanks for taking my second and third concern seriously and improving text and image quality.
1) However, when it comes to the main part of my criticism (concern #1) I do not see much improvement. In your answer you do not address my question for novelty.
2) In Fig 6 and in the text, according to your description, both data sets were measured on a CFRP plate. You keep comparing "CFRP plate vs. CFRP plate" or "CFRP composite vs. CFRP composite". This is very confusing. Why not call them sample A and sample B?
3) Eqaution (5) is missing an equal sign (=).
4) According to your Fig. 9, the correction factor makes the results worse, because the squares coincide much better with the solid line than the triangles, correct?
Author Response
Dear Sir;
Please find a file attached herewith.
Best wishes,
K.H.Im
